# GSVA: Gradient-Based Sparse Voxel Attacks on Point Cloud Object Detection

## Abstract

Point cloud object detection is crucial for a variety of applications, including autonomous driving and robotics. Voxel-based representation for 3D point clouds has drawn significant attention due to their efficiency and effectiveness. Recent studies have revealed the vulnerability of deep learning models to adversarial attacks, while considerably little attention is paid to the robustness of voxel-based point cloud object detectors. Existing adversarial attacks on the point cloud data involve generating fake obstacles, removing objects or producing fake predictions. Despite the demonstrated success, these approaches have three limitations. First, manipulating point data, which was originally designed for point-based representation, is inapplicable to voxel-based representation. Second, existing works that modified points in the hold scene led to redundant perturbations. Third, the evaluation primarily performed on small-scale datasets, such as KITTI, does not scale well. To address these limitations, we propose a gradient-based sparse voxel attack (GSVA) algorithm for voxel-based 3D point cloud object detectors. Two novel frameworks, i.e., re-voxelization-based voxel attack framework and light voxel attack framework, successfully modify voxel-based representation instead of raw points. In addition to KITTI, extensive experiments on large-scale datasets including nuScenes and Waymo Open Dataset demonstrate the favorable attack performance (with mAP decrease by $86.2\% \sim 99.5\%$) and the slight perturbation costs (with modification rate of $3.5\% \sim 30.6\%$) of our sparse attack algorithm.

## 1 Introduction

Point cloud 3D detection is crucial for accurate identification and understanding of objects in various fields, including autonomous driving and robotics Guo et al. (2021). However, processing the sparsity and disorder of LiDAR point cloud data poses a challenge. To address this, deep learning-based methods, including point-based Shi et al. (2019); Yang et al. (2020), voxel-based Zhou & Tuzel (2018); Lang et al. (2019); Yan et al. (2018); Yin et al. (2021); Shi et al. (2020b) and projection-based Chen et al. (2017b); Ku et al. (2018) have been developed. Voxel-based detection methods have been widely investigated Alaba & Ball (2022); Fernandes et al. (2021) and used in industry-level systems like Apollo Bai (2023) and Autoware Aut (2023) because of their better real-time performance Guo et al. (2021). However, recent studies have shown that deep learning models are vulnerable to adversarial attacks Szegedy et al. (2014); Goodfellow et al. (2015). Attackers can introduce subtle changes to the input data, causing incorrect predictions and failure of the system to identify critical objects. Therefore, understanding the potential vulnerabilities of point cloud detection algorithms to adversarial attacks is critical for enhancing the security and robustness of these algorithms.

In recent years, various methods of adversarial attacks against LiDAR-based 3D detection algorithms have been proposed. Cao et al. (2019) shot malicious laser pulses to autonomous vehicles to generate front-near fake obstacles. Cai et al. (2020) and Wang et al. (2021b) perturbed all the point clouds in a scene to make the car vanish by designing IOU-based attacks. Object insertion attacks have been proposed in Cao et al. (2019); Tu et al. (2020); Yang et al. (2021). Cao et al. (2019) and Tu et al. (2020) optimized an adversarial 3D mesh object to disappear the target obstacle or vehicle, and Yang et al. (2021) aimed at generating fake predictions of car for victim detector. Li et al. (2021) investigates the impact of manipulating the vehicle's trajectory with small perturbations to achieve a smooth and imperceptible attack. Based on our knowledge, the existing works have the following limitations: (1) **Focus on point-based representation**: They primarily concentrated on point-based

networks, which led to the lack of exploration of white-box attacks against voxel-based representation, due to the non-differentiability in the preprocessing phase. (2) **Redundant perturbation**: Related works Cai et al. (2020); Wang et al. (2021b) indiscriminately modified points of a hold scene, which resulted in unnecessary perturbation and harm of stealthiness. (3) **Limited exploration**: Most of the works evaluated their schemes on the early dataset KITTI Geiger et al. (2012), which has limited scale and diversity, leading to a lack of sufficient exploration on larger-scale datasets such as nuScenes Caesar et al. (2020) and completely no exploration on finely annotated datasets like the Waymo Open Dataset (WOD) Sun et al. (2020) to validate the generalizability and robustness of existing works.

Targeting the above limitations, we proposed a gradient-based sparse voxel attack (GSVA) scheme for voxel-based 3D LiDAR object detectors, as shown in Figure 1. Confronted with the non-differentiability of the preprocessing process, we design a re-voxelization-based voxel attack framework and light voxel attack framework, that target the modification on the voxelized feature, instead of the raw points. To boost stealthiness and enhance attack performance, we achieve sparse modification by employing an attention mechanism based on gradient information and detection boxes to identify points with higher attack effectiveness. We summarize our contributions as follows:

- We propose two novel frameworks for white-box attacks against voxel-based detection networks to settle the non-differentiability preprocessing. Note that we are the first to add adversarial perturbation to voxelized features to obtain adversarial examples.
- For the first time, we explore a sparse attack scheme against point cloud detectors when performing a hole scene perturbation, which locates the point cloud with higher attack efficiency by applying an attention mechanism.
- By applying the proposed frameworks, we validate our scheme and various classic algorithms by extensive experiments on KITTI, nuScenes and WOD, demonstrating higher attack performance with lower perturbation costs of our sparse attack. Note that nuScenes lacks thorough investigation for adversarial resilience, and WOD remained utterly unexplored, whose results are validated by official challenges [1].

In the following, a review of adversarial attacks on 2D images and 3D space is presented in Section 2, followed by a detailed introduction of our white-box attack framework for voxel-based networks and our attention-based technique for completing sparse attacks in Section 3. Our method is validated and analyzed experimentally in Section 4, and a conclusion is provided in Section 5.

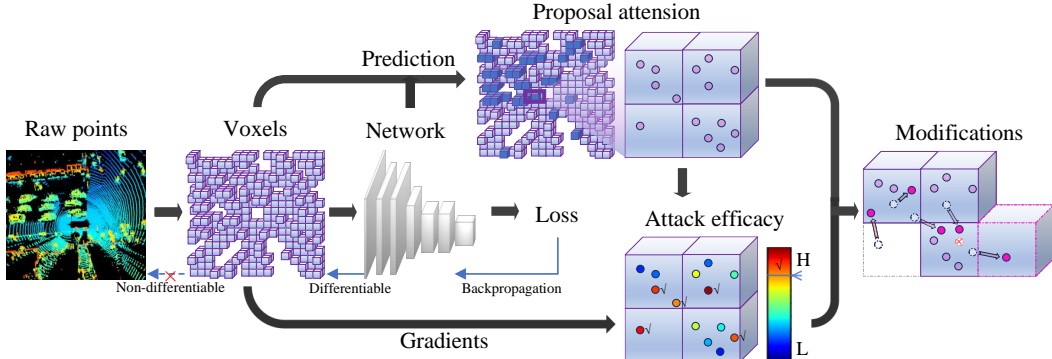

Figure 1: Overview of our sparse voxel attack algorithm. During the modification, points may move across voxels, which leads to removing voxels, creating new voxels or discarding points due to the restriction of the maximum number of points in each voxel (set as 5 in the given example).

## 2 Related work

In this section, we introduce the related work by first presenting the classical works on image adversarial attacks, then discuss the existing attack methods on 3D space, where we first introduce the works targeted on LiDAR-based point cloud detection methods, then briefly summarize works on multi-sensor fusion detectors, and conclude our findings in the end.

[1] https://waymo.com/open/challenges/

## 2.1 Attacks on 2D images

Image adversarial attacks have become a prominent research topic in the field of computer vision due to their potential security implications Akhtar et al. (2021). These attacks involve manipulating images with imperceptible perturbations to deceive deep learning models into making incorrect predictions. Early methods focused on generating adversarial examples, with the Fast Gradient Sign Method (FGSM) Goodfellow et al. (2015) being a prominent approach, which computes the perturbations based on the gradients of the loss function with respect to the input image. Iterative methods like Momentum Iterative FGSM (MI-FGSM) Dong et al. (2018) and Projected Gradient Descent (PGD) Madry et al. (2019) have since been proposed, which iteratively update the perturbations to maximize the model's vulnerability to adversarial inputs. adversarial attacks can be broadly classified into white-box attacks Xie et al. (2017); Moosavi-Dezfooli et al. (2016), and black-box attacks Brendel et al. (2017); Chen et al. (2017a) based on whether the attacker has complete or limited knowledge of the targeted model, respectively. Among these two types, white-box attacks are considered more fundamental and powerful as they enable the attacker to understand the inner workings of the targeted model and exploit its weaknesses directly. Adversarial attacks have also expanded to more complex scenarios, such as object detection Zhang & Wang (2019); Xie et al. (2017) and segmentation Arnab et al. (2018); He et al. (2020).

## 2.2 Attacks on 3D space

**Attacking LiDAR-based detectors.** Cao et al. (2019) explored the strategy of controlling the spoofed points to fool the machine learning model by sensor attack. Cai et al. (2020); Wang et al. (2021b) achieved a vanishing attack of the cat category by optimizing the perturbation through an efficient dual loss, including an adversarial loss based on the IOU (Intersection over Union) between the predicted and ground true boxes and a perturbation loss. Li et al. (2021) developed a method based on polynomial trajectory perturbation to achieve a smooth and imperceptible attack to lower the performance of the state-of-the-art detectors. Recent work Liu et al. (2023) applied a differentiable proxy and a probing algorithm to maximize the execution time of the detection system. The following several works performed object insertion attack by applying LiDAR simulator lid (2019); Möller & Trumbore (2005) on mesh object to obtain the simulated points. Tu et al. (2020) optimized a universal object on rooftops of malicious vehicles to hide the vehicles from LiDAR-based detectors. Yang et al. (2021) generated a roadside adversarial object that produce fake vehicles and investigated the reactions of an autonomous driving system in a simulated environment. Cao et al. (2019) proposed a differentiable proxy function for feature aggregation in the preprocessing phase to deceive an industry-level auto vehicle detection system with an undetectable adversarial object.

**Attacking multi-sensor fusion detectors.** Park et al. (2021b) and Park et al. (2021a) modified the image channel while Wang et al. (2021a) modified the LiDAR channel, which validated that sensor fusion models could be vulnerable when one of the input sources was manipulated. Xiong et al. (2021) and Liu et al. (2021) utilized generative adversarial networks (GANs) Goodfellow et al. (2014) framework to generate adversarial examples to attack image and LiDAR perception systems. Cao et al. (2021); Tu et al. (2022); Abdelfattah et al. (2021) optimized an adversarial 3D mesh object and placed it on the top of the vehicle or at the roadside to cause a target vehicle to disappear or produce a fake obstacle.

Based on the above summary, we find that related works Cai et al. (2020); Wang et al. (2021b); Tu et al. (2020); Yang et al. (2021); Li et al. (2021) only explored white-box attack against point-based detectors, while that of voxel-based detection methods were missing because of its non-differentiable preprocessing. Note that, the differentiable approximation functions employed in Cao et al. (2019; 2021); Liu et al. (2023) were only suitable for projection-based LiDAR detectors. In this paper, we proposed two novel frameworks to drive the development of the white-box attack against voxel-based LiDAR detectors.

# 3  Voxel attack

To accelerate the preprocessing pipelines of voxel-based LiDAR detectors, voxelization is typically implemented in C++[2] or Numba[34], where the gradients could not back-propagate from voxels to the input point clouds. Such non-differentiability leads to researchers relying on black-box attacks on voxel-based LiDAR detection networks to exploit their vulnerability Cai et al. (2020); Wang et al. (2021b); Tu et al. (2020); Yang et al. (2021); Li et al. (2021). To address the challenge, this paper focuses the modifications on voxel features for the first time (as illustrated in Figure 1) by introducing the two voxel attack frameworks (summarized in Algorithm 1 and Algorithm 2) and achieving sparse attacks.

| **Algorithm 1:** Re-voxelization-based voxel attack framework | **Algorithm 2:** Light voxel attack framework |
|---|---|
| **Input:** Raw input point clouds $\boldsymbol{p}$; number of iterations $T$; network weight $\theta$; ground true label $\boldsymbol{y}_{gt}$. | **Input:** Raw input point clouds $\boldsymbol{p}$; number of iterations $T$; network weight $\theta$; ground true label $\boldsymbol{y}_{gt}$. |
| **Output:** Adversarial point clouds $\boldsymbol{p}_{adv}$. | **Output:** Adversarial point clouds $\boldsymbol{p}_{adv}$. |
| 1: $[\boldsymbol{v_0}, \boldsymbol{c_0}, \boldsymbol{n_0}] = h(\boldsymbol{p})$ // Initilization | 1: $[\boldsymbol{v_0}, \boldsymbol{c_0}, \boldsymbol{n_0}] = h(\boldsymbol{p})$ // Initilization |
| 2: **for** $i = 0$ to $T - 1$ **do** | 2: **for** $i = 0$ to $T - 1$ **do** |
| 3: $\quad \boldsymbol{g}_i = \nabla_{\boldsymbol{v}_i} L(f(\theta, \boldsymbol{v}_i, \boldsymbol{c}_i, \boldsymbol{n}_i), \boldsymbol{y}_{gt})$ | 3: $\quad \boldsymbol{g}_i = \nabla_{\boldsymbol{v}_i} L(f(\theta, \boldsymbol{v}_i, \boldsymbol{c}_0, \boldsymbol{n}_0), \boldsymbol{y}_{gt})$ |
| 4: $\quad$ Get $\boldsymbol{\delta}_i$ by applying $\boldsymbol{g}_i$ with customized algorithm | 4: $\quad$ Get $\boldsymbol{\delta}_i$ by applying $\boldsymbol{g}_i$ with customized algorithm |
| 5: $\quad \boldsymbol{v}_i = \boldsymbol{v}_i + \boldsymbol{\delta}_i$ | 5: $\quad \boldsymbol{v}_i = \boldsymbol{v}_i + \boldsymbol{\delta}_i$ |
| 6: $\quad [\boldsymbol{v}_{i+1}, \boldsymbol{c}_{i+1}, \boldsymbol{n}_{i+1}] = h(h'(\boldsymbol{v}_i, \boldsymbol{n}_i))$ // Re-voxelization | 6: **end for** |
| 7: **end for** | 7: **return** $\boldsymbol{p}_{adv} = h'(\boldsymbol{v}_T, \boldsymbol{n}_0)$ |
| 8: **return** $\boldsymbol{p}_{adv} = h'(\boldsymbol{v}_T, \boldsymbol{n}_T)$ | |

## 3.1  Re-voxelization-based voxel attack framework

Voxelization is a common technique in point cloud processing that regularizes point cloud information, making it easier for computer vision and deep learning tasks to process. It first divides the point cloud scene into small and uniform cubes known as voxels. Each point within the detection range is then mapped to a corresponding voxel based on its coordinates. The information within each voxel is then aggregated and gives three outputs: 1) the voxelized feature $\boldsymbol{v} \in \mathbb{R}^{N_v, N_m, N_f}$, where $N_v, N_m, N_f$ is the number of voxel grids, the maximum number of points in a grid and the dimension of features, including the information of each point cloud data such as the location of three orthogonal axes, intensity, and timestamps, 2) the 3D position coordinate of each voxel $\boldsymbol{c} \in \mathbb{R}^{N_v, 3}$, and 3) the number of points within the voxels $\boldsymbol{n} \in \mathbb{R}^{N_v}$. It can be formulated as:

$$[\boldsymbol{v}, \boldsymbol{c}, \boldsymbol{n}] = h(\boldsymbol{p}), \tag{1}$$

where $\boldsymbol{p}$ is the raw points within the detection range, $h(\cdot)$ is the voxelization process. Since the gradient information of $\boldsymbol{v}$ is available, the adversarial perturbations are conducted on the position coordinates along the x, y, and z axes of $\boldsymbol{v}$. For better generality, we present our framework using an iterative manner (commonly used, such as PGD). Using subscript $i$ as the iteration index, the iterative procedure for the proposed re-voxelization-based voxel attack framework can be formalized as follows:

$$\boldsymbol{\delta}_i = \varepsilon \cdot sign(\boldsymbol{g}_i) = \varepsilon \cdot sign(\nabla_{\boldsymbol{v}_i} L(f(\theta, \boldsymbol{v}_i, \boldsymbol{c}_i, \boldsymbol{n}_i), \boldsymbol{y}_{gt})), \tag{2}$$

$$\boldsymbol{v}_{i+1} = \boldsymbol{v}_i + \boldsymbol{\delta}_i, \tag{3}$$

where $\theta$ is the weight parameters of target network $f(\cdot)$, $\boldsymbol{y}_{gt}$ is the ground true vector of the input data, $L$ is the loss function of the target network, $\boldsymbol{g}_i$ is gradients obtained by the partial derivative of $L$ with respect to the input $\boldsymbol{v}_i$, and $\varepsilon$ is the hyperparameter that adjusts the magnitude of the perturbation vector during iterations. Then the re-voxelization is performed to update $[\boldsymbol{v}, \boldsymbol{c}, \boldsymbol{n}]$:

$$[\boldsymbol{v}_{i+1}, \boldsymbol{c}_{i+1}, \boldsymbol{n}_{i+1}] = h(h'(\boldsymbol{v}_i, \boldsymbol{n}_i)). \tag{4}$$

---

[2]https://github.com/open-mmlab/OpenPCDet
[3]https://github.com/open-mmlab/mmdetection3d
[4]https://github.com/poodarchu/Det3D

where $h'(\cdot)$ is a devoxelization function that reassembles the point information in the voxel grids into point clouds. We denote the maximum number of iterations as $T$, then the resulting adversarial point clouds can be obtained by: $\boldsymbol{p}_{adv} = h'(\boldsymbol{v}_T, \boldsymbol{n}_T)$.

## 3.2 Light voxel attack framework

The above framework updates $[\boldsymbol{v}, \boldsymbol{c}, \boldsymbol{n}]$ every time by re-voxelization before the modified voxels are fed into networks to get gradients in the next iteration. For example, the re-voxelization is executed $T$ times for MI-FGSM, while it can be $T \times N_{GV}$ times for its variant algorithm like VMI-FGSM Wang & He (2021), where $N_{GV}$ is the number of samples used to obtain Gradient Variance, which could be very time-consuming. Based on this consideration, we further proposed a subtle and interesting framework, named light voxel attack framework, to speed up the attacks, which only performed voxelization once as depicted in Algorithm 2 line 1. Then $\boldsymbol{c}$ and $\boldsymbol{n}$ stayed the same during back-propagation (line 3) in each iteration and $\boldsymbol{v}$ only updated by $\boldsymbol{\delta}$ (line 5). The resulting adversarial points are obtained by $\boldsymbol{p}_{adv} = h'(\boldsymbol{v}_T, \boldsymbol{n}_0)$.

The essential difference between the two voxel attack frameworks is that, in the re-voxelization-based framework, the extracted voxel features, hash table for sparse convolution Yan et al. (2018); Yin et al. (2021); Shi et al. (2020b) and index tensor for scattering the pillar features back to 2D pseudo-image Lang et al. (2019) are iteratively readjusted, while in the light version framework, its hash table and index tensor are unchanged during iterations. Experiments in Section 4 showed that the light framework achieved comparable results and sometimes better performance for particular attack algorithms.

## 3.3 Sparse voxel attack

A global perturbation applied to all points may generate redundant perturbations with little attack efficacy. We employ the above frameworks to address this issue and introduce a sparse attack method that uses an attention technique based on the 3D boxes to locate and modify the most effective points.

Given a set of predicted boxes, we first obtain a mask matrix $\boldsymbol{M_B}$ that indicates whether a point of voxelized grid is inside one of the boxes:

$$M_B^{x,y,z} = \begin{cases} 1, & \text{if } v^{x,y,z} \text{ locates in predicted boxes,} \\ 0, & \text{otherwise,} \end{cases} \tag{5}$$

where $v^{x,y,z}$ and $M_B^{x,y,z}$ are point in $\boldsymbol{v}$ with position coordinate $(x, y, z)$ and its corresponding mask value in $\boldsymbol{M_B}$, respectively. Then we evaluate the attack efficacy $\boldsymbol{e}$ for each point according to their gradients, and then obtain a set of gradients $\boldsymbol{G}^\tau$, whose elements with the top $\tau$ attack efficacy $\boldsymbol{e}$ and located in predicted boxes simultaneously:

$$\boldsymbol{e} = \left\{ e^{x,y,z} = \|g^x, g^y, g^z\| \mid \forall g^{x,y,z} \right\}, \tag{6}$$

$$\boldsymbol{G}^\tau = sort(\boldsymbol{g} \cdot \boldsymbol{M_B}, \boldsymbol{e} \cdot \boldsymbol{M_B}, \tau), \tag{7}$$

where $\tau$ is a hyperparameter that adjusts the modification rate, $g^x$, $g^y$ and $g^z$ are the three components of $g_b^{x,y,z}$ along the x, y, and z axes, respectively, and $\|\cdot\|$ denotes the $L_1$ norm. After that, we can obtain a sparse mask matrix $\boldsymbol{M_S}$ and update the global perturbation $\boldsymbol{\delta}$ with $\boldsymbol{M_S}$ to obtain the resulting sparse perturbation for an iteration by $\boldsymbol{\delta} \Leftarrow \boldsymbol{\delta} \cdot \boldsymbol{M_S}$:

$$M_S^{x,y,z} = \begin{cases} 1, & \text{if } g^{x,y,z} \in \boldsymbol{G}^\tau, \\ 0, & \text{otherwise.} \end{cases} \tag{8}$$

# 4 Experiments

In this section, we will first introduce the basic information of the dataset, the experimental setting in detail and metrics for evaluating attacking performance and stealthiness, then the experimental comparison are reported, followed by the analysis of the optimal parameter, various sparse strategy, transferability and the visualized results.

Table 1: Evaluation results on the validation dataset of KITTI with PV-RCNN.

| Victim | Algorithm | Easy↓ (*RCE*↑) | Moderate↓ (*RCE*↑) | Hard↓ (*RCE*↑) | MR↓ | $D_{norm}$↓ | $D_{chamfer}$↓ |
|---|---|---|---|---|---|---|---|
| | Clean | 93.1 | 89.0 | 88.5 | - | - | - |
| | Clean$_R$ | 93.1 | 89.1 | 88.6 | - | - | - |
| | FGSM$_l$ | 5.1 (94.5) | 3.8 (95.8) | 3.6 (95.9) | 100.0% | $1.6 \times 10^4$ | 0.406 |
| | IOU-ADV$_l$ | 8.8 (90.6) | 4.7 (94.7) | 4.4 (95.0) | 99.1% [5] | $1.1 \times 10^4$ | 0.422 |
| PV-RCNN | PGD$_l$ | 16.4 (82.3) | 8.5 (90.4) | 8.6 (90.3) | 100.0% | $8.3 \times 10^3$ | 0.326 |
| | PGD$_R$ | 30.0 (67.7) | 23.2 (73.9) | 22.7 (74.3) | 100.0% | $6.5 \times 10^3$ | 0.253 |
| | MI-FGSM$_l$ | 10.9 (88.3) | 5.3 (94.0) | 6.0 (93.2) | 100.0% | $1.0 \times 10^4$ | 0.389 |
| | MI-FGSM$_R$ | 15.0 (83.8) | 8.5 (90.5) | 8.6 (90.3) | 100.0% | $8.8 \times 10^3$ | 0.347 |
| | Average | 14.4 (84.6) | 9.0 (89.9) | 9.0 (89.8) | 99.8% | $1.0 \times 10^4$ | 0.357 |
| | **GSVA$_l$** | 3.5 (96.2) | 2.5 (97.2) | 2.0 (97.7) | **5.5%** | $\mathbf{1.0 \times 10^3}$ | **0.028** |
| | **GSVA$_R$** | **1.8 (98.1)** | **1.4 (98.5)** | **1.5 (98.3)** | 6.3% | $\mathbf{1.0 \times 10^3}$ | 0.031 |

## 4.1 Datasets

**KITTI**: The KITTI dataset is a pioneering resource in autonomous driving research, containing 3712 training samples, 3769 validation samples and 7518 test samples. The evaluation metric we employed is the Average Precision (AP) with 40 recall positions at Easy, Moderate and Hard levels of difficulty with a rotated IoU threshold 0.7 for cars.

**nuScenes**: The nuScenes dataset emerges as a substantial and diverse dataset, consisting of over 1000 scenes, with 28k, 6k, and 6k annotated samples for the training, validation, and test sets, respectively. In the 3D detection task, the main evaluation metrics used are mean Average Precision (mAP) and nuScenes detection score (NDS). The mAP is calculated based on bird's-eye-view center distance thresholds instead of the standard box overlap, while NDS is a weighted average of mAP and other attribute metrics, including translation, scale, orientation, velocity, and other box attributes.

**Waymo Open Dataset**: Notable for its scale and complexity, WOD provides accurate and detailed annotations with rich sensor data. It contains 700, 150, and 150 sequences for training, validation, and testing, respectively, containing 64 lanes of Lidar points every 0.1s. The official 3D detection evaluation metrics include 3D bounding box mean average precision (mAP) and mAP weighted by heading accuracy (mAPH). Due to computational resource limitations, $6k$ samples in the validation dataset are randomly selected for experiments on WOD.

## 4.2 Experimental setup

To validate the proposed schemes, we apply various network architectures, including voxel-based, pillar-based and point-voxel-based networks, as our target networks. They are PV-RCNN Shi et al. (2020a), PointPillars Lang et al. (2019) and CenterPoint Yin et al. (2021) incorporating two widely used encoder architectures: VoxelNet Yan et al. (2018); Zhou & Tuzel (2018); Zhu et al. (2019) and PointPillars, denoted as CenterPoint-Voxel and CenterPoint-Pillar, respectively. For comparison purposes, we introduced FGSM, PGD, MI-FGSM, and IOU-based attack methods denoted as IOU-ADV. Specifically, IOU-ADV (used in Cai et al. (2020); Wang et al. (2021b); Tu et al. (2020)) aims to minimize the confidence and IoU of the relevant 3D candidates: $L = \sum_{y,s\in\gamma} -log(IoU(y^*, y))log(1 - s)$, where $\gamma$ is the set of box proposals, $y$ is a proposal with confidence score $s$ and $y^*$ is the ground true box. The maximum perturbation magnitude was set to 0.2 for these methods. For PGD and MI-FGSM, we set the number of iterations to 10, while IOU-ADV did not involve iterations (based on our experiments, it outperformed the iterative versions). In our method, we defined two hyperparameters, namely $(\varepsilon, \tau)$, the selection of their values is detailed in section 4.5.1. The perturbation truncation limit is set as 0.5 to prevent excessive modifications.

For brevity, we use subscript "R" to represent the employment of the re-voxelization voxel attack framework (such as PGD$_R$), while "l" is for the light version. Due to the page limitation, the results of pillar-based networks are reported in the Appendix.

---

[5]Very few samples remain unaltered for IOU-ADV as the scores of the predicted boxes are too low or there is no overlap between the predicted boxes and ground true boxes.

Table 2: Evaluation results on the validation dataset of nuScenes using CenterPoint-Voxel.

| Victim | Algorithm | mAP↓ (*RCE*↑) | NDS↓ (*RCE*↑) | MR↓ | $D_{norm}$↓ | $D_{chamfer}$↓ |
|---|---|---|---|---|---|---|
| | Clean | 59.6 | 66.8 | - | - | - |
| | Clean$_R$ | 59.6 | 66.8 | - | - | - |
| | FGSM$_l$ | 17.4 (70.8) | 34.3 (48.6) | 100.0% | $1.2 \times 10^5$ | 0.391 |
| | IOU-ADV$_l$ | 18.2 (69.5) | 34.7 (48.0) | 100.0% | $4.0 \times 10^4$ | 0.218 |
| | PGD$_l$ | 17.4 (70.8) | 36.3 (45.6) | 100.0% | $1.0 \times 10^5$ | 0.294 |
| | PGD$_R$ | 20.5 (65.6) | 38.9 (41.8) | 100.0% | $5.6 \times 10^4$ | 0.197 |
| | MI-FGSM$_l$ | 17.4 (70.8) | 34.2 (48.7) | 100.0% | $1.1 \times 10^5$ | 0.345 |
| | MI-FGSM$_R$ | 16.0 (73.2) | 33.7 (49.5) | 99.9% | $8.3 \times 10^4$ | 0.280 |
| | Average | 17.8 (70.1) | 35.4 (47.0) | 100.0% | $8.7 \times 10^4$ | 0.288 |
| | **GSVA**$_l$ | 8.2 (86.2) | 25.9 (61.1) | 20.6% | $5.1 \times 10^4$ | 0.126 |
| | **GSVA**$_R$ | **4.0 (93.3)** | **22.8 (65.8)** | **18.7%** | $\mathbf{3.5 \times 10^4}$ | **0.092** |

*Victim column (rotated): CenterPoint-Voxel*

Table 3: Evaluation results on the validation dataset of WOD using CenterPoint-Voxel.

| Victim | Algorithm | mAP↓ (*RCE*↑) | mAPH↓ (*RCE*↑) | MR↓ | $D_{norm}$↓ | $D_{chamfer}$↓ |
|---|---|---|---|---|---|---|
| | Clean | 70.1 | 68.4 | - | - | - |
| | Clean$_R$ | 70.7 | 68.5 | - | - | - |
| | FGSM$_l$ | 13.5 (80.7) | 13.1 (80.9) | 100.0% | $1.7 \times 10^5$ | 0.366 |
| | IOU-ADV$_l$ | 13.9 (80.2) | 13.4 (80.5) | 99.5% | $1.7 \times 10^5$ | 0.365 |
| | PGD$_l$ | 14.8 (78.9) | 14.3 (79.1) | 100.0% | $1.3 \times 10^5$ | 0.282 |
| | PGD$_R$ | 22.8 (67.4) | 21.9 (68.0) | 100.0% | $7.3 \times 10^4$ | 0.214 |
| | MI-FGSM$_l$ | 13.0 (81.4) | 12.6 (81.6) | 100.0% | $1.5 \times 10^5$ | 0.329 |
| | MI-FGSM$_R$ | 13.6 (80.6) | 13.1 (80.9) | 100.0% | $1.1 \times 10^5$ | 0.289 |
| | Average | 15.3 (78.2) | 14.7 (78.5) | 99.9% | $1.3 \times 10^5$ | 0.308 |
| | **GSVA**$_l$ | **1.9 (97.4)** | **1.8 (97.4)** | 7.9% | $2.6 \times 10^4$ | 0.040 |
| | **GSVA**$_R$ | 3.8 (94.5) | 3.8 (94.5) | **8.1%** | $\mathbf{2.1 \times 10^4}$ | **0.036** |

*Victim column (rotated): CenterPoint-Voxel*

## 4.3 Evaluation metrics

**Security Metrics**: To assess the security impact of different attack algorithms on the target detection networks, we use mAP, NDS, mAPH, and their *relative corruption error (RCE)* Dong et al. (2023) by measuring the percentage of performance drop as $RCE = (AP_{clean} - AP_{adv})/AP_{clean}$.

**Perturbation metrics**: We evaluate the sparsity of our algorithm's perturbation variables using the modification rate (MR) metric, which measures the proportion of the number of perturbed points to the total number of points. We employ regularized distance to measure the overall perturbation, ($L_1$ norm is used in this paper), denoted as $D_{norm}$. Furthermore, we apply the widely used chamfer distance Xiang et al. (2019); Wen et al. (2022) to measure the similarity or dissimilarity between two point sets, denoted as $D_{chamfer}$. Note that due to the voxelization process, some points may be discarded during the attack iterations. Therefore, we calculate perturbations for the voxelized points that are retained.

## 4.4 Evaluation Results

We introduced various network architectures and evaluated different algorithms on the KITTI, nuScenes and WOD, as presented in Tables 1, 2, 3, 5, 6 and 7 (see Appendix). Based on the experimental results, we make the following three observations. (1) *Voxelization on clean samples*. The use of voxelization on clean samples did not impact the detection results, indicating that the damage to detection performance was introduced by the attack algorithm, not voxelization itself. (2) *Light framework vs re-voxelization-based framework*. For voxel-based and point-voxel-based networks, attack schemes, like PGD and MI-FGSM, performed better when using the light framework, while the re-voxelization-based framework worked better for pillar-based networks. (3) *GSVA*. The proposed GSVA demonstrated less perturbation while achieving superior attack performance, for example, our method achieved $97.2\% \sim 99.2\%$ RCE of moderate difficulty on KITTI with much lower $D_{norm}$ and $D_{chamfer}$, while only modified less than 6.3% points. The proposed method achieved the most sparse modifications on KITTI and performed better with the re-voxelization-based voxel attack framework most of the time compared to the light framework.

## 4.5 Analysis

### 4.5.1 Optimal parameters

To optimize the performance of our algorithm, we carefully conducted a parameter-tuning process. To be specified, We defined appropriate ranges and intervals for $\{\varepsilon, \tau\}$: $\varepsilon \in \{0.1, 0.2, 0.3, 0.4, 0.5\}$, $\tau \in \{0.1, 0.2, 0.3, 0.4, 0.5\}$. To conduct extensive experiments and save computational costs, we conducted experiments on the training set of the mini-nuScenes dataset (with 323 samples) and randomly selected an identical number of samples in the validation dataset of KITTI and WOD. Results are shown in Figure 2 and Figure 5 (see Appendix). As shown in the Figures, increasing $\varepsilon$ or $\tau$ results in improved attack performance, but also leads to larger perturbations, and the attacking performance gradually converges as the iterations progress. We selected parameter combinations that balance attack performance and perturbation magnitude.

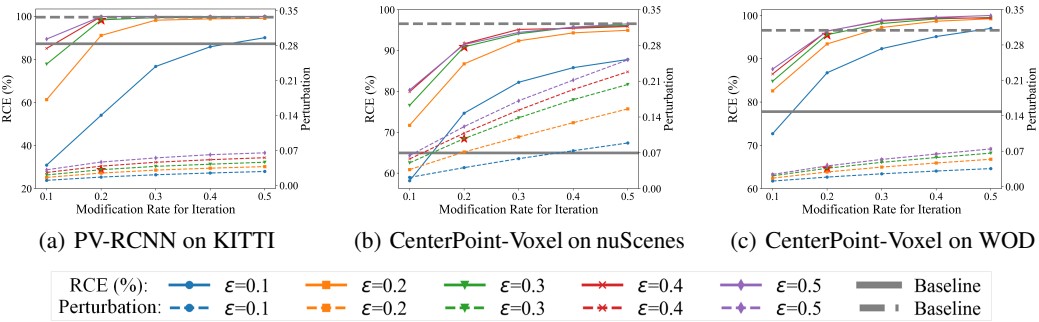

(a) PV-RCNN on KITTI     (b) CenterPoint-Voxel on nuScenes     (c) CenterPoint-Voxel on WOD

Figure 2: Experimental results for finding the optimal parameter setting for our algorithm on the mini version of KITTI, nuScenes and WOD. with PV-RCNN and CenterPoint-Voxel. Note that the gray lines represented the average results of the four baseline attack algorithms, i.e., FGSM$_l$, MI-FGSM$_R$, PGD$_R$ and IOU-ADV$_l$. The red asterisk symbols represented the parameter combinations we selected after considering a trade-off between attack performance and perturbation.

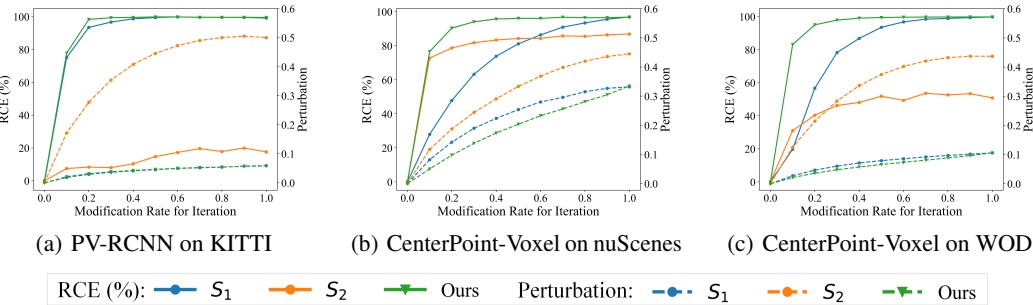

(a) PV-RCNN on KITTI     (b) CenterPoint-Voxel on nuScenes     (c) CenterPoint-Voxel on WOD

Figure 3: Comparison results for different strategies of sparse attack with PV-RCNN and CenterPoint-Voxel on the mini version of KITTI, nuScenes and WOD.

### 4.5.2 Sparse strategy comparison

To assessed the effectiveness of our sparse attack algorithm in locating key points, we introduced two different localization strategies, referred to as $S_1$ and $S_2$, for comparison. In strategy $S_1$, the attack efficacy was calculated in an opposite way at each iteration: $\boldsymbol{e} \Leftarrow -\boldsymbol{e}$, while in strategy $S_2$, we searched the key points outside the predicted boxes: $\boldsymbol{M_B} \Leftarrow \boldsymbol{1} - \boldsymbol{M_B}$. As shown in Figure 3 and Figure 6 (see Appendix), our method achieves the best performance with less perturbation magnitude and consistently outperforms $S_1$ in terms of attack performance and imperceptibility. Besides, while $S_2$ achieved some level of performance when $\tau$ was small, it failed to approach the performance of our method as $\tau$ increased. The above observations indicated the effectiveness of the designed attack efficacy and attention scheme.

### 4.5.3 Transferability

To further explore the effectiveness of our method in the black-box attack setting, we evaluated the transferability of our attack with various network architectures, including point-based (Point-RCNN

Table 4: Transferability ($RCE\uparrow$) of GSVA with various models on KITTI at easy, moderate and hard levels of difficulty.

| Source \ Target | Pv-RCNN | | | PointPillar | | | Point-RCNN | | | Voxel-RCNN | | |
|---|---|---|---|---|---|---|---|---|---|---|---|---|
| Pv-RCNN$_l$ | 96.2 | 97.2 | 97.7 | 48.2 | 57.0 | 62.5 | 74.4 | 81.7 | 84.3 | 100.0 | 100.0 | 100.0 |
| Pv-RCNN$_R$ | 98.1 | 98.5 | 98.3 | 59.4 | 65.9 | 70.4 | 84.2 | 89.9 | 90.8 | 90.5 | 93.7 | 94.0 |
| PointPillar$_l$ | 99.4 | 99.3 | 98.8 | 94.2 | 95.6 | 96.4 | 75.4 | 82.8 | 84.5 | 100.0 | 100.0 | 100.0 |
| PointPillar$_R$ | 99.5 | 99.3 | 98.8 | 98.6 | 98.1 | 98.4 | 73.6 | 80.4 | 81.7 | 92.2 | 94.6 | 94.0 |

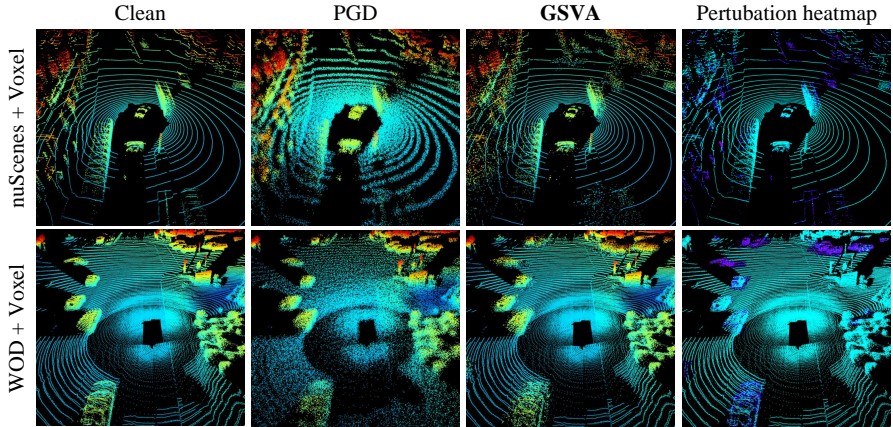

Figure 4: Visualization results of our method and PGD on nuScenes and WOD when attacking CenterPoint-Voxel. From left to right, the figures represent the results for a clean sample, PGD$_R$, GSVA$_R$, and a heatmap visualizing GSVA$_R$'s perturbation magnitude. The heat map gradient spans from cyan to purple, reflecting the range of perturbation magnitude from low to high, while the other images exhibit a color gradient from cyan to red, representing varying heights from low to high.

Shi et al. (2019)), voxel-based (Voxel-RCNN Deng et al. (2021)), pillar-based (PointPillar) and point-voxel-based (PV-RCNN) detectors. The quantitative results are shown in Table 4. As shown in Table 4, the proposed GSVA achieved remarkable performance under the transfer setting, leading to the average precision reduced by $48.2\% \sim 100\%$. The light voxel attack framework occasionally outperformed the re-voxelization-based framework.

### 4.5.4 Visualization

We compared our method with a representative baseline in Figure 4 and Figure 8 (see Appendix), which showed that PGD generated perturbations that were uniformly distributed across the entire scene that can be easily perceived by human eyes, while GSVA selectively modifies the points with high attack efficiency based on our sparse strategy, leading to remarkably better concealment of the adversarial examples. From the modification heatmap, we find that the perturbation magnitude of distant objects is larger than the nearby objects. This is an interesting result, and we consider the reason to be that: recognizing distant objects is more difficult, thus it attracts more attention for our sparse strategy and causes more modifications. Visualization for the resulting prediction are attached in the Appendix.

## 5 Conclusion

By identifying three limitations in the existing literature: the focus on point-based methods, the redundant perturbation and the limited exploration of larger-scale and finely annotated datasets, we proposed two novel voxel attack frameworks for white-box attacks and a sparse strategy to locate points with high attack efficacy. Extensive experiments were conducted on KITTI, nuScenes and WOD, validating higher attack performance with lower perturbation costs of the proposed scheme. In the future, it is highly worthwhile to develop more white-box attack methods for voxel-based detection networks by applying our voxel attack frameworks.

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

# A   Appendix

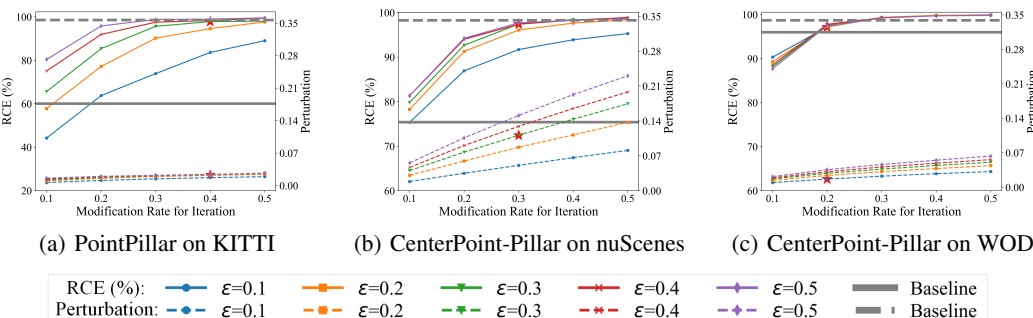

(a) PointPillar on KITTI  (b) CenterPoint-Pillar on nuScenes  (c) CenterPoint-Pillar on WOD

Figure 5: Experimental results for finding the optimal parameter values for our algorithm with pillar-based detectors (PointPillar and CenterPoint-Pillar) on the mini version of KITTI, nuScenes and WOD. Note that the gray lines represent the average results of the four baseline attack algorithms, including $FGSM_l$, $MI\text{-}FGSM_R$, $PGD_R$ and $IOU\text{-}ADV_l$. The red asterisk symbols represent the parameter combinations we selected after considering a trade-off between attack performance and perturbation magnitude.

Table 5: Evaluation results on the validation dataset of KITTI with PointPillar.

| Victim | Algorithm | Easy↓ ($RCE$↑) | Moderate↓ ($RCE$↑) | Hard↓ ($RCE$↑) | MR↓ | $D_{norm}$↓ | $D_{chamfer}$↓ |
|---|---|---|---|---|---|---|---|
| | Clean | 93.0 | 87.3 | 84.5 | - | - | - |
| | $Clean_R$ | 93.0 | 87.3 | 84.5 | - | - | - |
| | $FGSM_l$ | 68.5 (26.3) | 51.7 (40.7) | 45.3 (46.4) | 100.0% | $1.0 \times 10^4$ | 0.424 |
| | $IOU\text{-}ADV_l$ | 78.0 (16.1) | 63.6 (27.1) | 57.1 (32.4) | 97.3% | $1.0 \times 10^4$ | 0.413 |
| PointPillar | $PGD_l$ | 66.9 (28.1) | 49.1 (43.7) | 44.4 (47.4) | 100.0% | $5.5 \times 10^3$ | 0.247 |
| | $PGD_R$ | 10.4 (88.8) | 6.7 (92.3) | 5.9 (93.0) | 100.0% | $5.7 \times 10^3$ | 0.255 |
| | $MI\text{-}FGSM_l$ | 51.2 (44.9) | 38.5 (55.9) | 33.3 (60.6) | 99.9% | $7.2 \times 10^3$ | 0.310 |
| | $MI\text{-}FGSM_R$ | 15.3 (83.6) | 10.3 (88.2) | 9.0 (89.4) | 100.0% | $8.0 \times 10^3$ | 0.335 |
| | Average | 48.4 (48.0) | 36.7 (60.6) | 32.5 (65.1) | 99.5% | $7.8 \times 10^3$ | 0.331 |
| | **$GSVA_l$** | 5.4 (94.2) | 3.8 (95.6) | 3.1 (96.4) | 4.0% | $1.0 \times 10^3$ | 0.026 |
| | **$GSVA_R$** | **0.5 (99.5)** | **0.7 (99.2)** | **0.4 (99.5)** | **3.5%** | **$8.4 \times 10^2$** | **0.021** |

Table 6: Evaluation results on the validation dataset of nuScenes using CenterPoint-Pillar.

| Victim | Algorithm | mAP↓ (*RCE*↑) | NDS↓ (*RCE*↑) | MR↓ | $D_{norm}$↓ | $D_{chamfer}$↓ |
|---|---|---|---|---|---|---|
| CenterPoint-Voxel | Clean | 59.6 | 66.8 | - | - | - |
| | Clean$_R$ | 59.6 | 66.8 | - | - | - |
| | FGSM$_l$ | 17.4 (70.8) | 34.3 (48.6) | 100.0% | $1.2 \times 10^5$ | 0.391 |
| | IOU-ADV$_l$ | 18.2 (69.5) | 34.7 (48.0) | 100.0% | $4.0 \times 10^4$ | 0.218 |
| | PGD$_l$ | 17.4 (70.8) | 36.3 (45.6) | 100.0% | $1.0 \times 10^5$ | 0.294 |
| | PGD$_R$ | 20.5 (65.6) | 38.9 (41.8) | 100.0% | $5.6 \times 10^4$ | 0.197 |
| | MI-FGSM$_l$ | 17.4 (70.8) | 34.2 (48.7) | 100.0% | $1.1 \times 10^5$ | 0.345 |
| | MI-FGSM$_R$ | 16.0 (73.2) | 33.7 (49.5) | 99.9% | $8.3 \times 10^4$ | 0.280 |
| | Average | 17.8 (70.1) | 35.4 (47.0) | 100.0% | $8.7 \times 10^4$ | 0.288 |
| | **GSVA$_l$** | 8.2 (86.2) | 25.9 (61.1) | 20.6% | $5.1 \times 10^4$ | 0.126 |
| | **GSVA$_R$** | **4.0 (93.3)** | **22.8 (65.8)** | **18.7%** | $\mathbf{3.5 \times 10^4}$ | **0.092** |

Table 7: Evaluation results on the validation dataset of WOD using CenterPoint-Pillar.

| Victim | Algorithm | mAP↓ (*RCE*↑) | mAPH↓ (*RCE*↑) | MR↓ | $D_{norm}$↓ | $D_{chamfer}$↓ |
|---|---|---|---|---|---|---|
| CenterPoint-Pillar | Clean | 64.1 | 60.1 | - | - | - |
| | Clean$_R$ | 64.1 | 60.1 | - | - | - |
| | FGSM$_l$ | 4.9 (92.4) | 3.9 (93.6) | 100.0% | $6.7 \times 10^4$ | 0.427 |
| | IOU-ADV$_l$ | 7.8 (87.8) | 6.8 (88.7) | 99.5% | $6.7 \times 10^4$ | 0.424 |
| | PGD$_l$ | 3.7 (94.2) | 2.8 (95.3) | 100.0% | $3.4 \times 10^4$ | 0.243 |
| | PGD$_R$ | **0.1 (99.8)** | **0.1 (99.9)** | 100.0% | $3.4 \times 10^4$ | 0.255 |
| | MI-FGSM$_l$ | 2.0 (96.9) | 1.4 (97.7) | 100.0% | $4.8 \times 10^4$ | 0.317 |
| | MI-FGSM$_R$ | 0.2 (99.7) | 0.1 (99.8) | 100.0% | $4.8 \times 10^4$ | 0.338 |
| | Average | 3.1 (95.1) | 2.5 (95.8) | 99.9% | $5.0 \times 10^4$ | 0.334 |
| | **GSVA$_l$** | 5.2 (91.9) | 4.6 (92.3) | 5.1% | $2.5 \times 10^3$ | 0.018 |
| | **GSVA$_R$** | 3.0 (95.4) | 2.9 (95.2) | **4.6%** | $\mathbf{2.3 \times 10^3}$ | **0.016** |

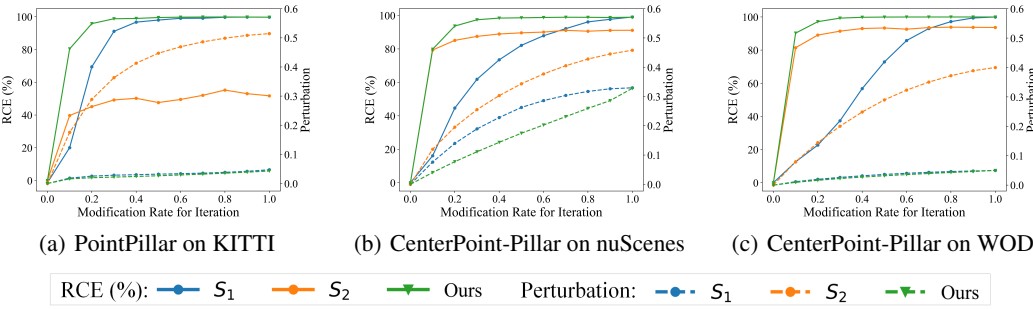

(a) PointPillar on KITTI   (b) CenterPoint-Pillar on nuScenes   (c) CenterPoint-Pillar on WOD

RCE (%): —●— $S_1$   —●— $S_2$   —▼— Ours    Perturbation: --●-- $S_1$   --●-- $S_2$   --▼-- Ours

Figure 6: Comparison results for different strategies of sparse attack with pillar-based detectors (PointPillar and CenterPoint-Pillar) on the mini version of KITTI, nuScenes and WOD.

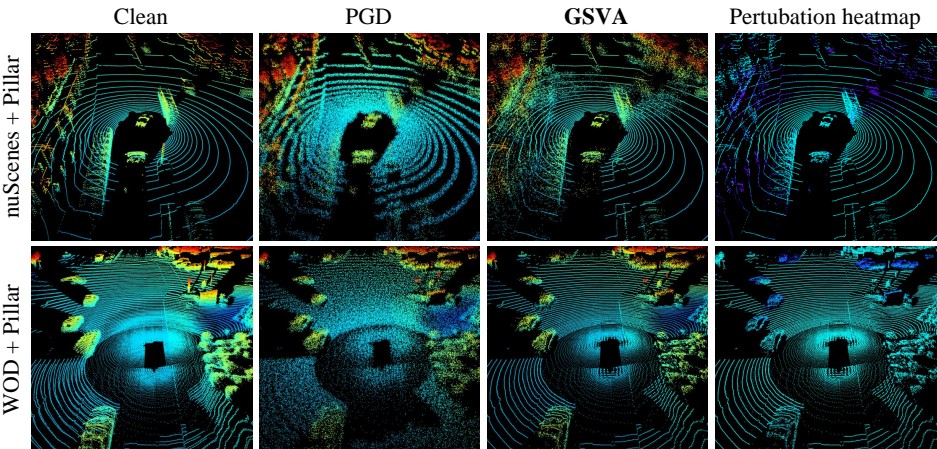

Figure 7: Visualization results of our method and PGD on nuScenes and WOD when attacking CenterPoint with PointPillar encoder architectures. From left to right, the figures represent the results for a clean sample, $PGD_R$, $GSVA_R$, and a heatmap visualizing $GSVA_R$'s perturbation magnitude. The heat map gradient spans from cyan to purple, reflecting the range of perturbation magnitude from low to high, while the other images exhibit a color gradient from cyan to red, representing varying heights from low to high.

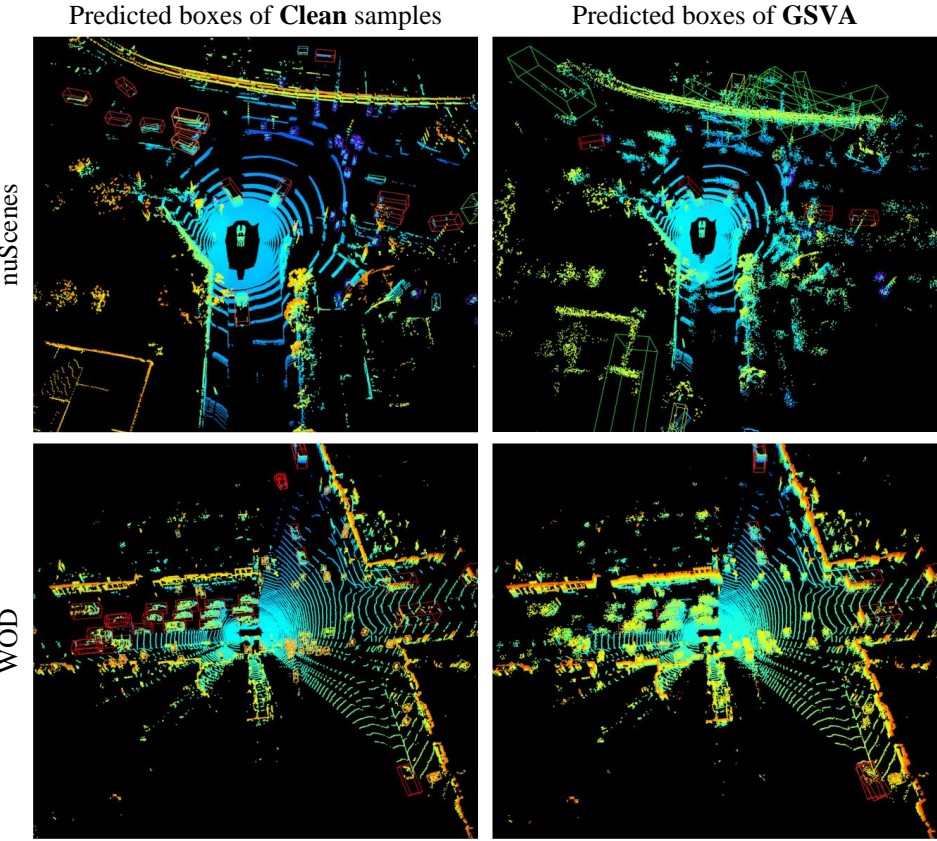

Figure 8: Visualized prediction results of the clean sample and adversarial examples obtained by our method on CenterPoint-Voxel. It demonstrates that the proposed GSVA can conduct fake prediction attacks and vanishing attacks simultaneously.

