# OpenReview forum: "GSVA: Gradient-Based Sparse Voxel Attacks \\ on Point Cloud Object Detection"
_ICLR.cc/2024/Conference — ICLR 2024 Conference Withdrawn Submission_

### Official Review · Reviewer_KepK · 2023-10-27

**Soundness:** 2 fair
**Presentation:** 2 fair
**Contribution:** 2 fair
**Rating:** 3
**Confidence:** 4

**Summary:**

This paper studies the problem of attacks on point cloud object detection and presents Gradient-based sparse Voxel Attacks (GSVA). Two attack frameworks, namely re-voxelization-based and light voxel attack, are proposed to modify voxel-baed representation. Experimental results are reported for three datasets: KITTI, nuScenes, and Waymo Open.

**Strengths:**

1. The proposed sparse voxel attack in Sec. 3.3 has some novelty in that is based on an attention technique that works on voxel-based representation.
2. The reported experimental results have demonstrated the effectiveness of the proposed GSVA and its superiority to more conventional attack methods such as FGSM and PGD.

**Weaknesses:**

1. This work is largely implementation-based and does not contain much theory. In terms of technical depth, it is not a good match with ICLR 's emphasis on theoretical foundation of learning representation. This work might be more appropriate for application-oriented conferences such as WACV and ACMMM.
2. The benchmark methods used in this study do not seem to represent the current SOTA on point cloud attacks. For example, ref. [1] [2] and likely there are more recently published works since then. A more thorough literature survey is needed.
3. In any security systems, attack and defense are two sides of the same coin. I understand this paper is about attacks on point cloud. But without any discussion about the related attack detection problem on the defense side, I am worried about the scientific soundness.

[1] Liu, Daizong, and Wei Hu. "Imperceptible transfer attack and defense on 3d point cloud classification." IEEE Transactions on Pattern Analysis and Machine Intelligence 45, no. 4 (2022): 4727-4746.
[2] Tao, Yunbo, Daizong Liu, Pan Zhou, Yulai Xie, Wei Du, and Wei Hu. "3DHacker: Spectrum-based Decision Boundary Generation for Hard-label 3D Point Cloud Attack." In Proceedings of the IEEE/CVF International Conference on Computer Vision, pp. 14340-14350. 2023.

**Questions:**

1. What are the motivation and advantages of voxel-based attacks compared to raw point-based ones?
2. Why sparsity? And how do you define sparse voxel attacks? It is difficult to appreciate the significance of sparsity in the current description of Sec. 3.3

---

### Official Review · Reviewer_Ck9c · 2023-10-31

**Soundness:** 3 good
**Presentation:** 2 fair
**Contribution:** 3 good
**Rating:** 6
**Confidence:** 3

**Summary:**

The paper proposes two frameworks for white-box attacks on voxel-based object detection networks. Specifically, there is a heavy (relative?) re-voxelization-based framework and a lightweight framework. The proposed method also applies a sparse voxel attack strategy based on an attention mechanism. The experiments conducted on KITTI, nuScenes, and WOD show great performance.

Overall, the proposed method is simple yet effective. More clear explanations or experiments may be needed for better understanding.

**Strengths:**

- The proposed method is simple but effective as shown in experiments on large-scale voxel-based point cloud detection networks.

- Experiments shown on large-scale lidar-based point cloud datasets are promising.

- The idea proposed in the paper could potentially be used in tasks other than object detection network attacks.

**Weaknesses:**

- Some arguments and experiment results may need further explanation. For example,
    - Some intuitive explanations or experiments are appreciated when doing parameter tuning in Figure 2 and 5.
    - What information does Table 4 give? Why do hard difficulty level experiments have the highest RCE?

- The paper argues that the proposed method can handle large-scale point cloud data. However, the authors should give more details about what benefits/advantages the proposed method gains when using large-scale point cloud data.

- The cost of using the proposed method should be provided for better understanding.

- The authors could add some comparisons of perturbation maps of other methods in Figure 4. In this way, the authors are able to prove the argument that the proposed method uses fewer perturbations.

- Overall, the writing of the paper could be improved. More details of the implementation could be provided for reproducibility.

**Questions:**

My main concern lies in some unclear experiments and arguments. I hope the authors can provide more details. Overall, the proposed method is simple and generalizable. Please see my detailed review above.

---

### Official Review · Reviewer_D947 · 2023-11-02

**Soundness:** 2 fair
**Presentation:** 1 poor
**Contribution:** 2 fair
**Rating:** 3
**Confidence:** 4

**Summary:**

Object detection using point clouds is essential for several fields, such as autonomous vehicles and robotics. The voxel-based approach to represent 3D point clouds has been favored for its efficiency and effectiveness. However, there is a growing awareness of the susceptibility of deep learning models to adversarial attacks, and the robustness of voxel-based point cloud object detectors hasn't been thoroughly investigated. The current adversarial strategies against point cloud data typically entail creating phantom obstacles, erasing objects, or inducing false predictions. Although effective, these methods face three key challenges. Firstly, they are designed for manipulating point-based representations and are not suitable for voxel-based ones. Secondly, altering points across the entire scene leads to unnecessary and excessive perturbations. Thirdly, the assessments are mostly done on smaller datasets like KITTI, which don't necessarily extrapolate well to larger scales. To overcome these issues, the authors introduce a new gradient-based sparse voxel attack (GSVA) algorithm tailored for voxel-based 3D point cloud object detectors. They have developed two innovative frameworks: the re-voxelization-based voxel attack framework and the light voxel attack framework, which specifically target voxel-based representations rather than the original point data. The paper'ss extensive testing on larger-scale datasets, such as nuScenes and Waymo Open Dataset, in addition to KITTI, showcases the strength of their sparse attack algorithm.

**Strengths:**

1. The motivation of the paper is valid. The gradient flow is indeed an issue in adversarial machine learning in 3D.
2. The authors tried to conduct a sufficient evaluation to back their methods.

**Weaknesses:**

1. The presentation of this paper is not very clear.
2. To me the re-voxelization attack does not make sense, the gradient is not accurate as every time the results of the voxelization will be different
3. It seems the authors only rely on PGD-styled attacks on all/most points and measure the overall mAP or NDS as the evaluation metrics. This is problematic in self-driving scenarios as the attack surface is artificial which is not applicable in the real world.

**Questions:**

Please refer to the weakness section. I need a very detailed explanation of the re-voxelization attack in how it solves the gradient obfuscation problem.